# A Novel and Simple Exercise Test Parameter to Assess Responsiveness to Cardiac Resynchronization Therapy

**DOI:** 10.3390/diagnostics10110920

**Published:** 2020-11-09

**Authors:** Alina-Ramona Cozlac, Lucian Petrescu, Simina Crisan, Constantin Tudor Luca, Cristina Vacarescu, Caius Glad Streian, Mihai-Andrei Lazar, Andra Gurgu, Angela Dragomir, Emilia Violeta Goanta, Radu Vatasescu, Kandara Mohamed Chahine, Ciprian Rachieru, Dragos Cozma

**Affiliations:** 1Cardiology Department, “Victor Babes” University of Medicine and Pharmacy, 2 Eftimie Murgu Sq., 300041 Timisoara, Romania; nicoramo_alina@yahoo.com (A.-R.C.); costiluca67@yahoo.com (C.T.L.); vacarescucristina@yahoo.com (C.V.); cstr100@gmail.com (C.G.S.); mihai88us@yahoo.com (M.-A.L.); gurgu_andra@yahoo.com (A.G.); ema.goanta@yahoo.com (E.V.G.); chahinek.med@hotmail.com (K.M.C.); ciprianrachieru@yahoo.com (C.R.); dragoscozma@gmail.com (D.C.); 2Cardiology Department, Institute of Cardiovascular Diseases Timisoara, 13A Gheorghe Adam Street, 300310 Timisoara, Romania; ange.dragomir@gmail.com; 3Department 4, Cardio-Thoracic Pathology, “Carol Davila” University of Medicine and Pharmacy, Dionisie Lupu Street, no. 37, Sector 2, 020021 Bucharest, Romania; radu_vatasescu@yahoo.com; 4Cardiology Department, Clinical Emergency Hospital Bucharest, 8 Calea Floreasca, 014461 Bucharest, Romania; 5Internal Medicine Department, County Emergency Hospital, 5 Gheorghe Dima Street, 300079 Timisoara, Romania

**Keywords:** cardiac resynchronization therapy, exercise test, heart rate recovery index, responder, non-responder

## Abstract

This study assessed the value of heart rate recovery index (HRRI), a new parameter of an exercise test, as the predictor of response to cardiac resynchronization therapy (CRT). *Methods*: Consecutive patients receiving a CRT device were followed up after implantation and every 6 months. An effort test (ET) was quantified by minimum heart rate/maximum heart rate, as well as acceleration and deceleration times. HRRI was calculated as the ratio between acceleration and deceleration time (AT/DT) and compared to outcome. We used logistic regression to assess the predictive value of HRRI for responders and non-responders to CRT. The area under the curve (AUC) was computed to distinguish between positive and negative outcomes. *Results*: A total of 109 patients (74 men, mean age 63.3 ± 9.8 years) were analyzed; permanent long-term fusion CRT pacing was possible in 65 patients. Patients were assigned to two groups: responders and non-responders (98/11 patients). During a mean follow-up of 36 months, 545 ETs were performed. HRRI was significantly higher in responders versus non-responders (3.16 ± 2 vs. 1.4 ± 0.5, *p* < 0.001). The optimal cutoff value for HRRI as a predictor of CRT response was 1.51 (area under the receiver operating characteristic (ROC) curve = 0.844). Responders had significant left-ventricular (LV) reverse remodeling (LV end-diastolic volume = 240 ± 90 mL vs. 217 ± 89 mL, *p* < 0.001) and higher LV ejection fraction (26 ± 5.8% vs. 35 ± 8.7%, *p* < 0.001). *Conclusions*: HRRI computation during routine ET is useful for the evaluation of responsiveness to CRT.

## 1. Introduction

Although current guidelines recommend systematic follow-up after cardiac resynchronization therapy (CRT), the optimization of device programming and control of capture is not sufficiently studied. About 30% of patients were shown to respond poorly to CRT, which may be due in part to poor follow-up after device implantation [1,2].

The exercise test (ET) is a noninvasive, low-cost investigation used for monitoring patients with CRT [3,4]. Several studies have shown that heart rate recovery after exercise cessation (HRR) is an independent prognostic marker for cardiovascular morbidity and all-cause mortality in healthy adult patients with cardiovascular disease and with diabetes mellitus [5,6,7,8,9,10,11,12]. In patients with heart failure, a blunted HRR, defined as a decrease ≤12 bpm, indicates lower exercise tolerance and more severe heart failure [13]. HRR computation, however, is difficult and time-consuming [6].

The aim of this study was to assess the association between ET-derived HRR and the response to CRT. We hypothesize that ET may be practical and applicable to study by analyzing the time ratio between the acceleration and deceleration of heart rate. Thus, a simpler and comprehensible tool may be developed to rapidly assess CRT patients.

## 2. Materials and Methods

The study population included consecutive patients with CRT indication, heart failure New York Heart Association (NYHA) class II–IV, left-ventricular ejection fraction (LVEF) ≤35%, QRS complex ≥130 ms, and optimal pharmacological treatment during the 3 months prior to CRT. Patients were excluded if any of the following was present: acute coronary syndrome during the preceding 3 months, permanent atrial fibrillation, severe comorbidities (e.g., renal, lung or liver failure, cerebral insufficiency, or terminal cancer), or noncardiac diseases that limit physical activity (e.g., orthopedic conditions).

### 2.1. Strategy Implantation

Direct percutaneous puncture of the left subclavian vein without initial incision was performed. An LV lead was placed in the target vein of the lateral, anterolateral, posterior, or posterolateral frame of the coronary sinus (CS), followed by incision and LV lead fixation; then, a subcutaneous pocket was formed, and the right atrium (RA) lead was placed at the end of the procedure. In patients with unfavorable CS anatomy, if endocardial pacing was not possible, the procedure was aborted and rescheduled after 1 month; if the second attempt was unsuccessful, the epicardial LV lead was surgically placed through mini-thoracotomy with implantation of a dual-chamber pacemaker. 

Clinical evaluation of patients included assessment of symptoms, NYHA functional class, and revaluated medication and co morbidities.

### 2.2. ECG Evaluation

A standard 12-lead electrocardiogram (ECG) (25 mm/s) was recorded at every follow-up visit. Serial follow-up ECGs were compared to the baseline ECG. Analyzed parameters included value of QRS axis and duration, presence of Q wave or a QS aspect V5, V6, which is indicative of the left lateral pacing, Rs/rS aspect in V1 and V2 with or without Q wave lead DI and aVL showing LV predominant pacing.

### 2.3. Transthoracic Echocardiography (TTE)

TTE was performed before and the day after CRT, as well as during follow-up at 3 month and 6 month intervals, using a Vivid E9 echocardiograph (GE Health Medical, Milwaukee, WI, USA) and a 2.5 MHz transducer. Complete TTE evaluation was performed in all patients [14]. Standard echocardiographic measurements were assessed according to echocardiography guidelines: left-ventricular end-diastolic diameter (LVEDD), left-ventricular end-diastolic volume (LVEDV), left-ventricular end-systolic volume (LVESV), left-ventricular ejection fraction (LVEF), left-atrial volume (LAV), and left-atrium diameter (LAd). Evaluation of dyssynchrony parameters involved assessment of interventricular, intraventricular, asynchronism, and septal flash.

### 2.4. Exercise Test (ET)

An ET was performed every 6 months after CRT among patients who were hemodynamically stable and without history of hospitalization for heart failure in the previous month. The cycloergometric ET was performed using a GE exercise system and the Bruce protocol with an automatic 25 W increase in the workload at each 2 min exercise stage. Blood pressure (BP), 12-lead ECG, and active interrogation of implantable cardiac devices were permanently recorded throughout the ET. Exercise capacity was measured in metabolic equivalents of task at peak exercise (METs). Patients were assessed by ET in sinus rhythm; all patients with atrial fibrillation (AF) at the time of evaluation were converted to sinus rhythm and rescheduled for ET. CRT assessment during ET included minimum/maximum heart rate, acceleration and deceleration time, and a newly derived index, heart rate recovery index (HRRI). HRRI was defined as the ratio between acceleration and deceleration time (AT/DT; see Figure 1).

### 2.5. Device Programming

The initial interrogation of the cardiac devices was performed at the end of the implant procedure, 24 h after implantation, and at each follow-up visit. The following steps were performed in all patients after CRT: 12-lead ECG pacing on/off and complete interrogation. All pacemakers were initially programmed at a base heart rate of 60 bpm and maximum tracking rate (MTR) of 130 bpm. Check-up included spontaneous AV interval, percentage of ventricular pacing, LV/RV threshold, and adequate response of cardiac devices functioning during exercise. The AV interval was programmed individually to achieve the best fusion or biventricular capture.

### 2.6. Response to CRT

Response assessment to CRT was based on the following criteria [14,15]:
-Clinical response to CRT, defined as improvement in NYHA functional class, ET duration, and workload.-Echocardiographic response (defined as >5% increase in LVEF, 15% decrease in LV end-systolic/diastolic volume, and decreased mitral regurgitation degree).-Assessment of outcomes: number of hospitalizations due to worsening heart failure, all-cause mortality, and morbidity.

All subjects gave their informed consent for inclusion before they participated in the study. The study was conducted in accordance with the Declaration of Helsinki, and the protocol was approved by the Ethics Committee of our institute (number 1622/26.03.2014).

### 2.7. Statistical Analysis

Statistical analysis was performed using IBM SPSS Statistics23 software. Data are summarized as percentage ± standard error, mean ± standard deviation if normally distributed, or median ± interquartile range if non-normally distributed. Statistical tests were used to compare numerical or nominal variables, using a statistical significance threshold of 0.05. A *t-*test was used for continuous variables and a chi-square test was used for categorial variables.

## 3. Results

The analytic study group included 109 patients out of 122 initially enrolled (74 men, aged 63.3 ± 9.8 years). The drop-off rate of the study was approximately 10% due to either impossibility of ET (orthopedic conditions, advanced heart failure symptoms) or lack of consistent data (missing more than three follow-ups).

### 3.1. Baseline Demographic Data

Baseline demographic data are shown in Table 1. Patients received the following types of pacemakers: St Jude Medical, Sorin Group, Medtronic, Biotronik, and Boston Scientific. The AV interval was individually programmed with an AV paced beat of 154 ± 32.4 and an AV sensed beat of 118 ± 27 with more than 95% effective CRT pacing in all patients. Epicardial cardiosurgical implantation off LV lead was performed in three patients with unfavorable coronary sinus anatomy.

The mean follow-up period was 36 months (range 12–60 months); 755 medical visits were performed. NYHA functional class improvement by at least one class was observed in all responder group patients (Figure 1). Mitral regurgitation deceased in 55 patients from the responder group.

### 3.2. Results: ECG and Echocardiography

Main data from average baseline ECG constituted a QRS duration of 168.89 ± 30.4 ms and QRS axis of −18° ± 36°. Table 2 shows the baseline echocardiographic data.

LV reverse remodeling with an improvement in echocardiographic parameters (decrease in LVEDV (240 ± 90 vs. 217 ± 89, *p* < 0.001) and increased LVEF (26 ± 5.8 vs. 35 ± 8.7, *p* < 0.001)) was observed 6 months after CRT. An improvement in echocardiographic asynchronism parameters was noticed in 109 patients (Table 3).

### 3.3. Exercise Test Results

A total of 545 ETs were performed during follow-up visits. It should be mentioned that the non-responder patients performed the ET at the 6 month follow-up visit, whereas no ETs were performed at subsequent visits due to them being obviously irrelevant.

ET was performed using the Bruce protocol in all patients with a mean of 109 ± 54 Watts (7.26 ± 3.3 metabolic equivalents of task) and a peak heart rate of 105 ± 27. Atrial fibrillation occurred in 12 patients (11%) at the time of evaluation; thus, ET was rescheduled over 1 month due to specific management to obtain a sinus rhythm. Direct current cardioversion for persistent atrial fibrillation was needed in these seven patients to achieve a sinus rhythm.

DT was significantly shorter in responder versus non-responder patients (125.86 ± 77.4 ms vs. 241.68 ± 116.6 ms, *p* < 0.001). HRRI was statistically significantly higher in responder versus non-responder patients (3.16 ± 2 vs. 1.4 ± 0.5, independent sample *t*-test, 95% confidence interval (CI), *p* < 0.001; Table 4, Figure 2).

Main comparative data for ET parameters are presented in Table 4.

The new parameter studied, HRRI, predicted susceptibility to the CRT response (area under the receiver operating characteristic (ROC) curve = 0.844, Figure 3). The optimal cutoff value for HRRI as a predictor for susceptibility was 1.51 (sensitivity 80%; specificity 80%). For other variables (METs, basal HR, maximum HR, and Watt), the model was not statistically significant, and the area under the ROC curve was smaller than that for HRRI.

### 3.4. Device Programming

Chronotropic incompetence was noted in five patients; these patients were not included in the ROC analysis, due to artificial “pseudonormal” behavior of the HR histogram (Figure 4) after reprogramming of the rate–response function of the CRT device. Maximum tracking rate was exceeded in 22 patients, whereas loss of capture by shortening physiological PR interval (16 patients) was observed as the main cause of loss of LV capture or inadequate pacing at ET. After the exercise test, CRT optimization was performed in 10 patients by increasing MTR at 145 beats/min, and dynamic AV interval was reprogrammed in 20 patients. Hospitalization due to worsening heart failure was needed for 31 patients (28%). By the end of the study, seven patients were lost (in five patients, non-sudden cardiac death occurred; two patients were lost to follow-up).

## 4. Discussion

Heart failure management in special populations is complex [16]. CRT fine-tuning and evaluation are time-consuming and sometimes subjective (e.g., NYHA class). Previous studies indicated that ET is useful in CRT device optimization, and the present study adds to this information by presenting a simpler and more comprehensible interpretation of heart failure evaluation during mid-term follow-up in a CRT population [3,4]. This study adds new information regarding the use of the ET as an important tool to improve CRT response. We introduced a novel parameter to simplify the ET heart rate curve analysis; responders and super responders presented an obvious steeper HRR compared to non-responder patients; thus, the AT/DT was statistically significantly higher (higher prolonged HRR time).

Other studies communicated similar results regarding an improvement in clinical parameters (NYHA class) and echocardiography parameters (increased LVEF, decreased LVESV) in the group of patients evaluated at 6 months after CRT [15,17].

HRR analysis was introduced to assess the relationship between recovery rate after reaching the maximum effort and all-cause morbidity and mortality; several investigators preferred a simple HRR calculation method by analyzing changes in heart rate from peak exercise to minute 1 or 2 of recovery [6,10]. A decrease in heart rate by 15–20 bpm in the first minute after cessation of exercise occurs in healthy population [8]. The HRR analysis method is complex and difficult to achieve due to several factors involved (e.g., motion artefacts, protocol used). Van de Vegte et al. studied HRR at 10, 20, 30, 40, and 50 s after cessation of exercise in 40,727 patients; the conclusion was that HRR calculated at 10 s is a superior predictor of outcome compared to HRR at other time intervals [18]. We add supplementary data to the usefulness of ET in the CRT population using a simpler parameter and easy-to-use method. Thomas et al. conducted a study on 37 patients with heart failure who underwent CRT [19]. The heart rate analysis was performed on the preimplant ET. They analyzed HRR at 30, 60, 90, and 120 s after cessation of exercise and heart rate deceleration HRD (30, 60, 90, and 120 s) using the arithmetic relationship between the maximum heart rate and the four-point recovery time. The conclusion of the study was that HRD correlates with the functional and echocardiographic response to CRT; the use of this parameter in association with classical criteria could bring new data in the evaluation of CRT patients [19].

Autonomic nervous system dysfunction is one of the mechanisms responsible for the vicious circle that worsens HF and leads to sudden cardiac death in these patients [20]. Several studies have shown that increased sympathetic system activity and decreased parasympathetic system activity were associated with an increased risk of sudden cardiac death and/or susceptibility to ventricular arrhythmias [21]. Various markers have been proposed to evaluate the function of the autonomic system, one of them being HRR at the end of physical exercise [22]. The present study is the first to evaluate a novel parameter HRRI using HR behavior after cessation of exercise in patients with CRT; baseline and maximal HR could not predict CRT response, but the analysis of ET graphs in responders versus non-responders showed a better outcome corelated with improved response in heart HRR. Thus, we can speculate an improvement in autonomic nervous system dysfunction, as the analysis of the HHRI parameter at 6 months after CRT showed a significantly higher value in patients with reverse remodeling. An explanation for this phenomenon was previously demonstrated by Gademan et al. [23]. In this paper, the authors revealed that CRT decreases the permanent neurohumoral activation by decreasing the involvement of the cardiac sympathetic afferent reflex [23]. Finally, a literature review concerning the response to CRT and implications for clinical outcome may include frailty as a novel independent factor that can be used to predict the clinical response to CRT in this special population [2,24,25,26].

### Limitations

In our study, the analyzed patients were either ischemic or not, using both fusion LV CRT pacing and true biventricular pacing [27]. Analysis using subgroups was not feasible as the population sample was too small. Another limitation is that the responder rate in our study was high, mainly due to a tight and tailored follow-up for each patient, obviously leading to a better outcome through refined device reprogramming and adaptive drug management.

One other limitation is that the HR and chronotropic response are affected by medication (beta-blockers, ivabradine) [28]; thus, values of HRRI and maximal HR should be interpreted accordingly. These are common limitations for patients with optimized medication for heart failure. However, the criteria for non-responders and responders did not interfere with medication.

## 5. Conclusions

HRRI is a novel tool in ET analysis and can be routinely introduced in the evaluation of patients to describe responsiveness to CRT. Optimizing cardiac devices and treatment may be improved using HRRI.

## Figures and Tables

**Figure 1 diagnostics-10-00920-f001:**
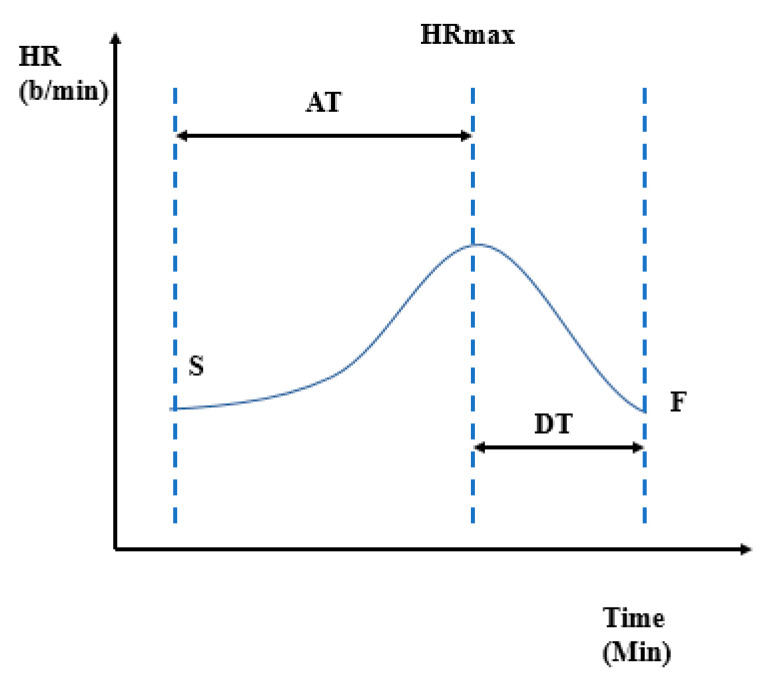
Graphic description of heart rate recovery index (HRRI) = acceleration time (AT)/deceleration time (DT). HR = heart rate; HRmax = maximum heart rate; S = start exercise; F = final exercise.

**Figure 2 diagnostics-10-00920-f002:**
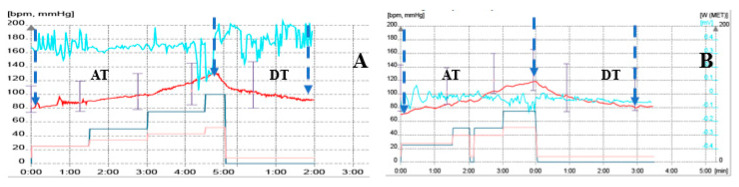
Example of HR diagram (red line) in a responder (**A**) and non-responder (**B**). HRRI is obviously better in patient A (HRRI = 2.5) versus patient B (HRRI = 1.33). AT = acceleration time; DT = deceleration time.

**Figure 3 diagnostics-10-00920-f003:**
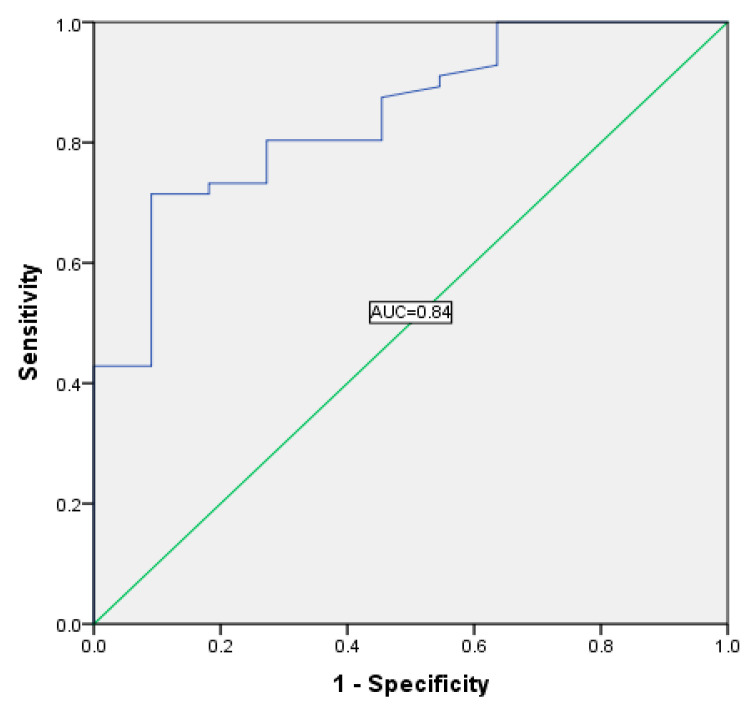
Receiver operating characteristic (ROC) curve for HRRI as a predictor of susceptibility to CRT response: area under the ROC = 0.844 for a value of HRRI = 1.51 representing cutoff point (sensitivity 80%; specificity 80%, *p* < 0.001).

**Figure 4 diagnostics-10-00920-f004:**
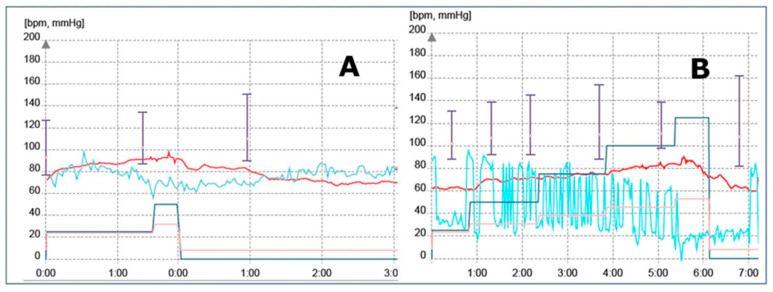
HR diagram (red line) in a patient with chronotropic incompetence (**A**) and redone ET after reprogramming rate–response function (**B**).

**Table 1 diagnostics-10-00920-t001:** Baseline demographic data.

All Patients (*n* = 109)	
Mean age (years)	63.32 ± 9.8
Male, *n* (%)	74 (68)
NYHA functional class n (%)	
II	9 (8)
III	83 (76)
IV	18 (16)
Hypertension, *n* (%)	85 (77.3)
Chronic kidney disease, *n* (%) *	61 (56)
Diabetes mellitus, *n* (%)	40 (36)
Dyslipidemia, *n* (%)	89 (81.3)
Medication, n (%)	
ACEI/ARB	99 (90.7)
Beta-blockers	90 (82.7)
Ivabradine	31 (28)
Diuretics	102 (93.3)
Aldosterone receptor antagonists	89 (81.3)
Digoxin	5 (4)

* Chronic kidney disease: reduction in creatinine clearance ˂90 mL/min. In our group, there were no patients with creatinine clearance value ˂30 mL/min. Abbreviations: ACEI, angiotensin-converting enzyme inhibitor; ARB, angiotensin receptor blockers.

**Table 2 diagnostics-10-00920-t002:** Baseline echocardiographic data.

	All Patients (*n* = 109)		Range
LVEDD (cm), mean ± SD	6.44 ± 0.95		4.5–8.9
LVEDV (mL), mean ± SD	240.32 ± 90.7		90–560
LVESV (mL), mean ± SD	176.45 ± 72.6		43–414
EF (%), mean ± SD	26.75 ± 5.8		15–40
LAV (mL), mean ± SD	120.73 ± 58.7		49–440
Valvular disease, n (%)	Mild	Moderate	Severe
Mitral regurgitation	64 (58.7)	24 (21.3)	22 (20)
Tricuspid regurgitation	85 (77.4)	22 (20)	3 (2.7)
Aortic regurgitation	24 (21.3)	2 (1.3)	0
Aortic stenosis	3 (2.7)	0	0
Asynchronism parameters	Interventricular asynchronism n (%)		
	50 (46)		
	Intraventricular asynchronism n (%)		
	42 (38)		
	Septal flash n (%)		
	58 (53)		

Abbreviations: EF, ejection fraction; LAV, left-atrial volume; LVEDD, left-ventricular end-diastolic diameter; LVEDV, left-ventricular end-diastolic volume; LVESV, left-ventricular end-systolic volume.

**Table 3 diagnostics-10-00920-t003:** Echocardiographic and electrical parameters before and after cardiac resynchronization therapy (CRT).

	Before CRT	After CRT	*p-*Value
LVEF (%)	26 ± 5.8	35 ± 8.7	<0.001
LVESV (mL)	176 ± 72	145 ± 73	<0.001
LVEDV (mL)	240 ± 90	217 ± 89	<0.001
LVEDD (cm)	6.4 ± 0.95	6.1 ± 1.1	<0.001
QRS interval (ms)	168.89 ± 30.4	134.22 ± 20.4	<0.001
QRS axis (°)	−18 ± 36	+38 ± 105	<0.001

Values are shown as means ± SD. Abbreviations: LVEDD, left-ventricular end-diastolic diameter; LVEDV, left-ventricular end-diastolic volume; LVEF, left-ventricular ejection fraction; LVESV, left-ventricular end-systolic volume.

**Table 4 diagnostics-10-00920-t004:** Comparison of exercise test (ET) parameters between responders and non-responders. MET, metabolic equivalents of task; HR, heart rate.

Parameter	Responders (*n* = 98)	Non-Responders (*n* = 11)	*p*-Value
METs	6.43 ± 1.2	6.04 ± 1.0	0.3
Watt	111.16 ± 32.6	102.27 ± 23.5	0.4
Basal HR	67.55 ± 10.2	63.64 ± 7.2	0.2
Maximum HR	100.55 ± 21.8	88.90 ± 10.7	0.09
AT	303.66 ± 125.9	303.7 ± 96.9	0.9
DT	125.86 ± 77.4	241.68 ± 116.6	<0.001
HRRI = AT/DT	3.16 ± 2.0	1.4 ± 0.5	0.007

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
