# Peer review of "A Novel and Simple Exercise Test Parameter to Assess Responsiveness to Cardiac Resynchronization Therapy"

_diagnostics, 2020, doi:10.3390/diagnostics10110920_

Round 1

Reviewer 1 Report

In this study, the authors examined the value of HRRI as a predictor of response to CRT. The article presented a new tool in the evaluation of the patients' responsiveness to CRT. The topic area is important. The authors contribute to the research literature in the area of this topic.

Having said this, some points/areas may be a matter of changes, reorganizing, and/or improving the text and its content:

The first-time mentioned that abbreviations should be written in full (lines 25, 26, 47).

Introduction – lines 47, 49, and 55 – dose numbers 1, 2, and 6 represent references number?

Materials and methods – line 86 – the authors wrote 'Complete ETT…' should it be TTE?

The authors should mention Figure 1 in subsection 2.4. Exercise time, not line 147, because it is not appropriate to place for Fig 1.

The authors did not mention Figure 3 anywhere in the manuscript. Please correct that.

The study's limitation is the number of patients in the responders (98) and non-responders (11) group. The authors should discuss that and how this could impair the results.

Author Response

Please find attached file for reviewer 1. 

Reviewer 2 Report

Dear Editor,

I read the work sent for review with great curiosity. The determinants of response to CRT are a real clinical problem worth developing and deepening.
The job is very interesting.
The purpose of the work is clearly defined. A sufficient group of patients was included in the study and the observation period was 3 months. It is worth extending the observation period, in order to reliably confirm the response to CRT therapy, the observation period should be extended to 6 months. what was your follow-up, please state clearly.
The methodology of the work has been precisely described, it is worth clarifying in the results in which number of patients was subject to subsequent procedures or cardiosurgical implantation.
In the ROC figures, please include the sensitivity, specificity, cut-off point and p value.

Figure 4 is hardly visible, it is necessary to improve the quality of the figure.

As the response to CRT and physical effort are influenced by many factors, it is worth discussing them in the discussion. Apart from individual comments, the discussion is interesting. Please quote the works below.

Modified frailty as a novel factor in predicting the response to cardiac resynchronization in the elderly population

Effect of cardiac resynchronization therapy
on left ventricular diastolic function: Implications for clinical
outcome.

Frailty as a predictor of negative outcomes after cardiac resynchronization therapy.
  Avoiding non-responders to cardiac resynchronization therapy: a practical guide  

Author Response

Please find attached answer to reviewer 2. 
